

# Sensitivity of aerosol optical depth trends using long term measurements of different sun-photometers

Angelos Karanikolas[1,2], Natalia Kouremeti[1], Julian Gröbner[1], Luca Egli[1], Stelios Kazadzis[1]

[1] World Radiation Center, Physikalisch-Meteorologisches Observatorium Davos (PMOD/WRC), Davos Dorf, Dorfstrasse 33 7260, Switzerland
[2] Physics department, ETH Zurich, Zurich, Franscini-Platz 5 8093, Switzerland

*Correspondence to*: Angelos Karanikolas (angelos.karanikolas@pmodwrc.com)

**Abstract.** This work aims to assess differences in the aerosol optical depth (AOD) trend estimations when using high quality AOD measurements from two different instruments with different technical characteristics, operational (e.g. measurement frequency), calibration and processing protocols. The different types of Sun photometers are the CIMEL that is part of AERONET (AErosol RObotic NETwork) and a precision Filter Radiometer (PFR), part of the Global Atmosphere Watch Precision Filter Radiometer network. The analysis operated for two wavelengths (500/501 nm and 870/862 nm for CIMEL/PFR) in Davos, Switzerland, for the period 2007-2019.

For the synchronous AOD measurements, more than 95% of the CIMEL-PFR AOD differences are within the WMO accepted limits, showing very good measurement agreement and homogeneity in calibration and post correction procedures. AOD trends per decade in AOD for Davos for the 13-year period of analysis were approximately -0.017 and -0.007 per decade for 501 nm and 862 nm (PFR), while the CIMEL-PFR trend differences have been found 0.0005 and 0.0003 respectively. The linear trend difference for 870/862 nm is larger than the linear fit standard error. When calculating monthly AODs using all PFR data (higher instrument frequency) and comparing them with the PFR measurements that are synchronous with CIMEL, the trend differences are smaller than the standard error. The trend differences are also larger than the trend uncertainty attributed to the instrument measurement uncertainty, with the exception of the comparison between the 2 PFR datasets (high and low frequency) at 862 nm. Finally, when calculating time-varying trends, they differ within their uncertainties.

## 1 Introduction

Aerosols from both anthropogenic and natural sources, are an important component regarding the study of atmospheric processes (Ginoux et al., 2012). They affect the Earth's energy budget and distribution by scattering and absorbing solar and terrestrial radiation. They also act as cloud condensation nuclei thus playing a crucial role in cloud formation and properties (Fan et al., 2016). Their effect on surface solar radiation is found to be a significant forcing of the climate (IPCC, 2021) and dominant on surface solar radiation variations for several decades (Wild, 2012; Wild et al., 2021). Surface solar radiation is important for its biological effects (mainly in the UV region) (Horneck 1995; Bais et al. 2018) and for solar energy applications



(Hou et al., 2021; Fountoulakis et al., 2021; Myers 2005). Their interactions with clouds are also important for radiative forcing attribution, climate modelling and weather forecasts (Rosenfeld et al., 2014; Glotfelty et al., 2019; Huang & Ding, 2021; Benedetti et al., 2018).

Aerosol concentration in the atmosphere is variable and changes according to the variability of sources and removal
mechanisms. Part of the aerosol load variability is due to changes on anthropogenic emissions (e.g. Lei et al., 2011; Zhao et al., 2017) but it can be influenced by natural factors such as volcanic activity (e.g. Vernier et al., 2011) and dust transport (e.g. Gkikas et al., 2022). For example, increasing load of Sahara dust is evident in Central Sahara region and decreasing in the Mediterranean and eastern parts of North Africa region (Logothetis et al., 2021). Therefore, their long-term trend study is important information for studying the climate variability and the solar radiation effects on ecosystems (Wetzel, 2003; Paul &
Gwynn-Jones, 2003; Edreira et al., 2020).

One of the most important parameters regarding aerosols is the aerosol optical depth (AOD). It is the parameter that describes the aerosol column direct effect on solar radiation and the most important aerosol-related parameter for Earth energy budget related studies (WMO, 2003). AOD is calculated through measuring the direct sun irradiance. There are various instrument networks like AERONET (Holben et al., 1998), the Global Atmospheric Watch-Precision Filter Radiometer (GAW-PFR)
(Kazadzis et al., 2018b) and the sky radiometer network (SKYNET) (Nakajima et al., 2020). AERONET is the largest network with around 400 stations in 50 countries all over the world and the instrument used is the CIMEL Sun photometer (Gregory, 2011). SKYNET covers approximately 100 stations, and its instrument is the PREDE-POM sun-sky radiometer (Nakajima et al., 2020). Most of the sites are in East Asia and Europe. GAW-PFR has 15 stations in all continents mainly at remote locations, aiming for long term measurements of background aerosol conditions. Its instrument is the PFR (Wehrli, 2000). Several studies
have examined the AOD measurement differences between different networks. The intercomparisons are either for short (campaign based) like the Filter Radiometer comparison (FRC) (Kazadzis et al., 2018a) or for longer periods (Estelles et al., 2012; Kim et al., 2008; Cuevas et al., 2019).

A key issue about AOD is its long-term variability as it is important for the study of the changes of aerosol contribution to the Earth-Atmosphere energy balance. AOD changes are associated with the variability of aerosol sources (Reddington et al.,
2016) and the atmospheric transport (Kumar et al., 2013). Several studies have investigated long-term trends of AOD from ground-based observation (e.g. Ningombam et al., 2019; Li et al., 2014; Nyeki et al., 2012) and from satellite observations (e.g. Cherian & Quaas, 2020; Guo et al., 2011). The accuracy of trends from ground-based instruments has a particular significance since the AOD ground-based observations are used for satellite validation (Kotrike et al., 2021; Wei et al., 2019; Ogunjobi & Awoleye, 2019; Ma et al., 2016; Xie et al., 2011), climate model validation (Mortier et al., 2020) and modeling
assimilation (Benedetti et al., 2018). There is lack of studies comparing the AOD trends derived from different instruments. Studying the AOD trend analysis limitations is an important step towards a better understanding and quantification of the trend uncertainties and reliability. A standard source of uncertainties is the instrument measurement uncertainty, which can be of the same magnitude as AOD, in low AOD locations. However, using two different instruments their calibration and post correcting differences can lead to large differences in AOD trend calculations from each one. Another source of uncertainty can possibly





be the measurement frequency of the instrument. Instruments with lower measurement frequency are likely to miss fluctuations of AOD during the day that might affect the results. As AOD is measured only without the presence of clouds, this can be amplified for days/months with few cloudless sky measurements. For the GAW-PFR network the measurement frequency is once every minute. AERONET has a default schedule of 1 measurement every 0.5 airmass interval for airmass > 2 and every 15 minutes for the rest of the day (Gregory, 2011).  In order to deal with the low availability of data, one can induce limits for

the number of measurements a month must contain in order the monthly data to be considered valid. In Nyeki et al. (2012) where data of PFRs are analysed, each month must contain at least 100 measurements. Extra limits are induced for daily and hourly values to 50 and 6 observations respectively. In Li et. al (2014), where CIMELs are used, the requirement of a valid monthly value is a minimum of 6 point-measurements per month. Given the limited geographic coverage of instruments that measure every minute and the high coverage of AERONET network, an interesting question is how significant the effect of

the measurement frequency can be on long-term variability detection of AOD and therefore how reliable trends can be calculated for most parts of the world.

Other source of uncertainty are the cloud related (cloud flagging) algorithms from different networks/instruments (Kazadzis et al., 2018a). Such differences can lead to a systematic overestimation of the AOD from algorithms that fail to deal with cloudy sky measurements and an underestimation of the AOD from algorithms that are too strict and characterize as "cloudy"

high and highly variable AOD cases (e.g. biomass burning aerosols, Gilles et al., 2019).

Finally, AOD averaging in order to calculate long term trends can be tricky. Due to cloud presence AOD is not a continuous measurement and the amount of data averaged from hourly up to monthly basis can differ spatiotemporally. In addition, since AOD measurements in a number of cases worldwide are not normally distributed, AOD averaging and calculating trends can influence the results of the analysis (O'Neil et al., 2000; Levy et al., 2009; Sayer & Knobelspiesse, 2019).

For this study, we use 13 years of parallel PFR and CIMEL timeseries in order to investigate their AOD differences and all the related uncertainties (calibration, algorithms, measurement frequency etc.) affecting their AOD trend calculation differences.

In the following section, we describe the location and the instruments used, followed by the methodology of the AOD intercomparison and finally the trend analysis methods. On Section 3 the results are presented and in Section 4 the conclusions.

**2 Instruments and methodology**

**2.1 Location and instrumentation**

The instruments used for this study are operated at PMOD/WRC in Davos. Davos is in a valley of a mountainous region in eastern Switzerland. The altitude of the station is 1590 m a.s.l. and there are no significant pollution sources nearby. However, aerosols can reach the area through long range transport from the various industrial and urban areas in Switzerland or

surrounding countries, but also from the Sahara Desert in cases of severe European dust episodes (Greilinger & Kasper-Giebl, 2021).





For this study, we use the PFR (N27) Sun photometer, which is part of the WMO AOD reference (Kazadzis et al., 2018b). Three CIMEL Sun photometers have been operated at Davos (2005-2018, 2018-2019 and 2019-present). The description for each instrument can be found at the following sections.

### 2.1.1 PFR

The Precision Filter Radiometer (described in Wehrli, 2000) is an automatic Sun photometer that measures the direct solar irradiance in 4 channels. It is mounted on a separate tracking system that continuously follows the motion of the Sun. Its channels extend from the near-UV to the near-IR and are centered on 368 nm, 412 nm, 501 nm and 862 nm. The radiation passes through interference filters in order to let only a narrow spectral region centered at these wavelengths reach the detector, which is a silicon photodiode. Their full-width-at-half-maximum (FWHM) bandwidth varies from 3 nm to 5 nm and its field-of-view angle (FOV) is approximately 2º in order to provide high confidence for the full solar tracking. It is a weatherproof instrument, highly protected from the outside conditions with its temperature kept constant at approximately 20ºC by an active Peltier system. It also has internal constant pressure of ~2 atm with dry nitrogen. Its filters are exposed to the solar radiation for 10 s every minute in order to measure direct solar irradiance. Each filter is in a constant position behind a different shutter so they can be exposed to the Sun the same moment.

Most of the PFRs are calibrated through comparison with the PFR reference triad. The triad is being compared regularly with specific PFRs, which are calibrated with the Langley Plot method (LP) (Shaw et al., 1973) in 2 high altitude locations (Mauna Loa in Hawaii-USA and Izaña in Tenerife-Spain) (Kazadzis et al., 2018b).

### 2.1.2 CIMEL

The CIMEL Sun photometer (described in Holben et al., 1998) is an automatic instrument with 2 axis robotic tracking system that measures the direct solar irradiance and diffuse sky radiance in the spectral range of 340 nm to 1640 nm for up to 10 wavelengths depending on its version. The CIMEL version used in this study has at least 8 interference filters centered at 340, 380, 440, 500, 675, 870, 940, and 1020 nm with 10 nm full-width-at-half-maximum (FWHM) bandwidth, except for 340 and 380 nm which have 2 and 4 nm FWHM, respectively. The irradiance is measured by a silicon detector, which measures each channel for 1 s and the filter wheel moves to the next until all channels are measured. The measurement sequence is repeated 3 times in a time interval of approximately 30 seconds. Its field-of-view angle (FOV) is 1.2º. It has a four-quadrant detector in order to improve the tracking of the Sun before the measurements by detecting the point with the maximum radiation intensity. The CIMEL Sun photometers are calibrated through LP at Mauna Loa station or with a calibration transfer from an instrument calibrated at Mauna Loa (Toledano et al., 2018) and their AOD retrieval algorithms are presented in Gilles et al., 2019).





## 2.2 Intercomparison methods and trend calculation

### 2.2.1 Measurement intercomparison

We compared AERONET/CIMEL and GAW/PFR AOD measurements and trends on the 2 channels that are directly comparable (CIMEL/PFR: 500/501 nm and 862/870 nm).

As mentioned, basic AOD differences among two different instruments are related with different calibration standards, technical and post correction differences and different AOD retrieval algorithms. In order to assess such differences, the World meteorological organization has defined the WMO criterion of traceability among instruments or Networks. It is defined as the number in percent of synchronous measurements that lie within ±(0.005±0.01/m), (m: air mass coefficient) (WMO, 2005). Traceability is established when more than 95% of such synchronous data are within those limits. Here we use the data of the period 2007-2019 and as synchronous we consider the measurements with maximum time difference of 30 seconds. The instruments were compared also in terms of the correlation of their monthly median values with the coefficient of determination ($R^2$) as criterion. Median values were selected instead of mean values because, as mentioned, AOD values do not follow a normal distribution and the data are non-continuous mainly due to clouds, but also due to shipment for calibration or instrument malfunction. We firstly compare the differences between the mean and the median values and then we use the medians for the rest of the comparisons.

In order to obtain more robust results, a monthly median is considered valid if there are at least 5 valid days of measurements for each month. Valid are considered days with at least 3 measurements during the day. In this study, a monthly median is the median of all valid daily medians during the month.

We also aim to assess the AOD differences due to the measurement frequency difference. The PFR is measuring every minute, while CIMEL less often. To isolate the effect of the measurement frequency, we compare 2 different PFR datasets. The first one, $PFR_{syn}$, is synchronous with the CIMEL data so it represents the CIMEL measurement frequency. The second one, $PFR_{hf,}$ is a much larger dataset that represents the PFR measurement frequency (1 min). The datasets and the corresponding number of measurements are shown in Table 1. The 2 instruments use different cloud screening algorithms. The synchronous datasets contain only measurements that are considered cloud-free according to both AERONET and GAW/PFR algorithms, while the $PFR_{syn}$ dataset is screened with the GAW/PFR algorithm only. The differences between the 2 algorithms showed no significant effect on the AOD. 93.8% of CIMEL data that are synchronous with PFR data were identified as cloud-free according to the PFR related algorithm. Keeping only this 93.8% of CIMEL and PFR synchronous data reduced the average AOD of both instruments by less than 0.002 at 500 nm and less than 0.0005 at 865 nm, pointing towards the conclusion that the cloud contamination effects on AOD are minimal.

From all comparisons only the valid days and months that are common to both instruments and also to synchronous and non-synchronous datasets are used for the trend analysis. The length of the whole period is 156 months, 131 of which include common CIMEL and PFR measurements (mainly due to absence of the CIMEL instruments for their calibration). The months satisfying the selection criteria for all datasets are 114.





Table 1: The number of measurements for the datasets and the time period used.

| Period | Dataset | N 500/501 nm | N 862/870 nm |
|---|---|---|---|
| 2007-2019 | CIMEL-PFR$_{syn}$ | 33197 | 33116 |
| 2007-2019 | PFR$_{hr}$ | 452281 | 452507 |

**2.2.2 Linear trends**

The linear trends were calculated with the least-squares linear regression (LSLR) method and their statistical significance has been identified by the non-parametric Mann-Kendall statistical test modified for autocorrelated data (Hamed & Rao, 1998). The timeseries for the trend calculation and detection were the de-seasonalized monthly medians of AOD for the period 2007-2019. The intra-annual cycle was calculated separately for each dataset from all medians for each month. To assess the trend
agreement, we compare the trend differences with the standard error of the fitting method (LSLR).

We also used the Monte Carlo method (Metropolis & Ulam, 1949) to examine whether the measurement uncertainty alone is capable of producing trend differences equal or larger than the observed. The uncertainty of the instruments at the selected channels is approximately 0.01 (Holben et. al., 1998; Kazadzis et al. 2020). By applying the Monte Carlo method to the AOD observations we calculate the uncertainty propagation of the measurement uncertainty to monthly AOD. Then we calculate the
propagation of the monthly AOD uncertainty to the AOD trends. In both cases, for each AOD measurement or monthly AOD median we generated 10000 normally distributed random values with the mean of the distribution being the corresponding observed AOD value (measurement or monthly median) and its standard deviation the corresponding uncertainty. The final output is 10000 AOD random timeseries for each dataset for which we calculate their trends. The standard deviation of those trends is the trend uncertainty due to the measurement uncertainty.

**2.2.3 Time-varying trends**

As long-term fluctuations of AOD are not necessarily monotonic or follow a linear trend for any given period (Streets et al., 2009) and static linear trends can also be sensitive to outliers (Bashiri & Moslemi, 2013), we examine how realistic is the assumption of the existence of linear trends for these timeseries. For this purpose we used the Dynamic Linear Modeling (DLM) method described in Laine et al. (2014) on the monthly median timeseries. This is a method for calculating trends that
vary through time using dynamic linear models (Petris et al., 2009) and Kalman filtering (Harvey, 1990). Any type of known periodicities and external forcings can be used as inputs in the model in order to be removed from the data points. For the DLM trend uncertainty quantification the Markov Chain Monte Carlo (MCMC) method is used (Gamerman, 2006). As





seasonal component we used only the calculated annual cycle, which is removed by the model using a harmonic function. The monthly median uncertainty is also a necessary input to the model for the calculation of the trend and its uncertainty.

The model output is monthly data including AOD change per month and its uncertainty, which here is scaled to AOD change per decade. The procedure was repeated for both synchronous and non-synchronous timeseries. The final trends are compared in relation to their 1σ uncertainty. They are also compared with LSLR trends.

# 3 Results

## 3.1 AOD data comparison

In this section we assess the AOD differences between monthly AOD calculated from mean AOD with monthly AOD calculated from median AOD and between the synchronous AOD data from CIMEL and PFRN27. Finally, we compare the CIMEL/PFR monthly AOD differences with the differences between the two PFR datasets representing different measurement frequencies (PFR$_{syn}$ and PFR$_{hf}$ Section 2.2.1).

The AOD intra-annual cycles calculated through the mean of AOD measurements differ to those calculated through the median 195 of AOD measurements (Table 2). The differences range between 0.0004 (December) to 0.023 (June) for 500 nm and 0.0001 (December) to 0.012 (June) for 865 nm. The differences correspond approximately from 1-2% (December) to 39% (June) of the 13-year average AOD for each month. Such differences can create trend differences (as discussed in Section 3.2).

Table 2: The difference between the intra-annual cycles calculated from AOD means and those calculated from AOD medians 200 for CIMEL and PFRN27. In both cases the intra-annual cycle for a month is the mean of all AOD medians/means of the month during 2007-2019 period.

| AOD intra-annual cycle difference median-mean$\times 10^{-3}$ | | | |
|---|---|---|---|
| month | CIMEL 500 nm | PFRN27 501 nm | CIMEL 870 nm | PFRN27 862 nm |
| 1 | -3.93 | -4.24 | -2.14 | -2.21 |
| 2 | -8.76 | -8.70 | -4.57 | -4.29 |
| 3 | -18.1 | -18 | -7.91 | -8.15 |
| 4 | -8.47 | -9.77 | -4.51 | -4.75 |
| 5 | -6.29 | -7.29 | -3.07 | -4.44 |
| 6 | -23.4 | -21.3 | -11.9 | -12.2 |
| 7 | -10.3 | -9.76 | -5.45 | -5.79 |
| 8 | -11.8 | -11.9 | -6.69 | -7.10 |
| 9 | -4.58 | -4.55 | -2.48 | -2.72 |
| 10 | -4.28 | -3.99 | -2.07 | -2.42 |
| 11 | -1.95 | -1.86 | -1.51 | -1.26 |
| 12 | -0.42 | -0.78 | -0.28 | -0.12 |





The instrument comparison showed a very good agreement for this 13-year period as 95.6% of the AOD differences in 500 nm and 98% in 865 nm are within the WMO limits. There is no evident time dependence of the AOD differences showing

good calibration consistency between CIMEL and PFR (Figure 1). There is also no evident dependence of the AOD differences with the air mass (Figure 2). The values outside the WMO limits (red lines in Figure 1 and 2) show the larger deviations at specific periods like the second half of 2019. The monthly AOD of the two instruments shows good correlation ($R^2 > 0.95$ for both channels) (Figure 3). In the Table 3 these results are summarized.

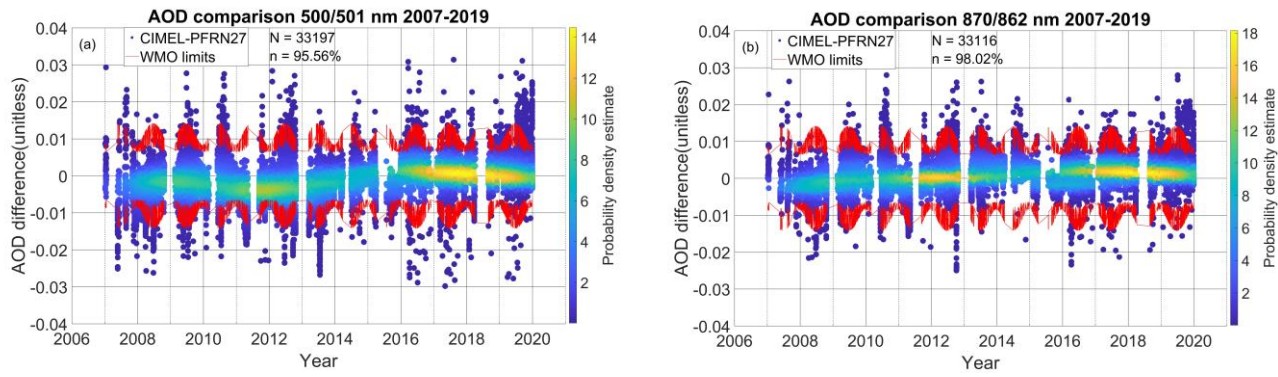


Figure 1: CIMEL-PFRN27 differences (blue points and light blue to yellow bands) and WMO limits (red lines) of synchronous AOD measurements with respect to time in years for 500/501 nm (a) and 870/862 nm (b) for the PFR/CIMEL. The colourbar corresponds to the density of the AOD difference data points.

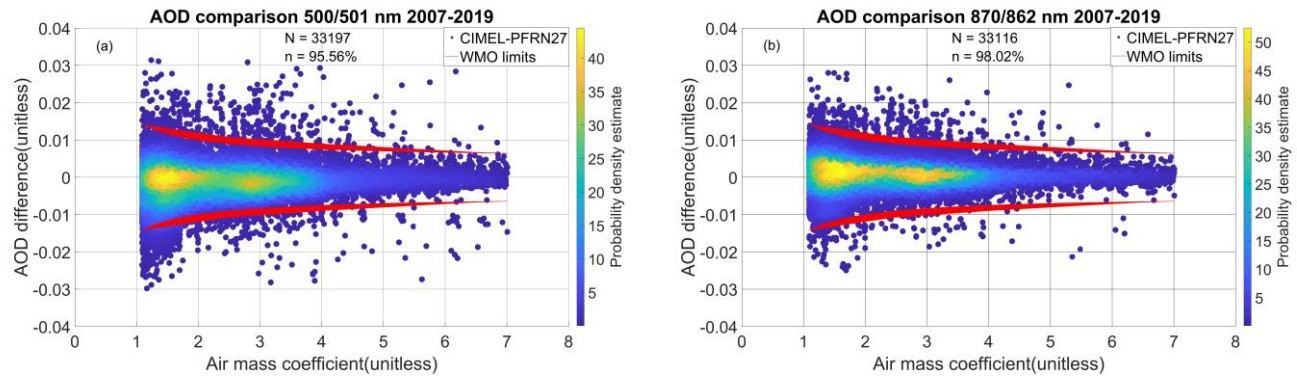


Figure 2: CIMEL-PFRN27 differences (blue points and light blue to yellow bands) and WMO limits (red lines) of synchronous AOD measurements with respect to air mass coefficient for 500/501 nm (left) and 870/862 nm (right) for the PFR/CIMEL. The colourbar corresponds to the density of the AOD difference data points.






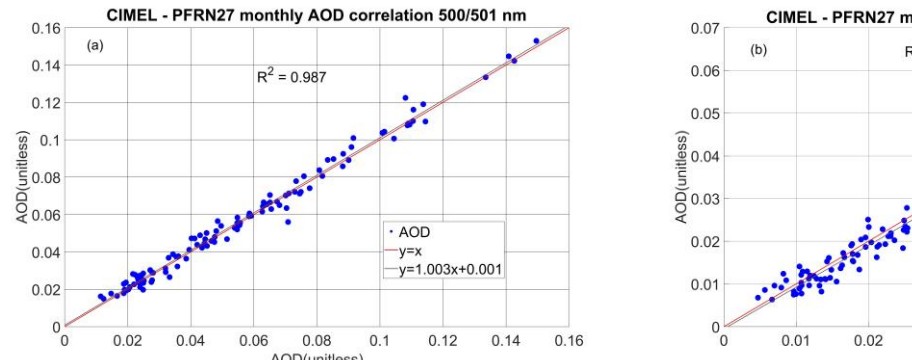

Figure 3: Scatter diagram of the monthly median AODs between CIMEL and PFRN27 501/500 nm (a) and 870/862 nm (b). The coefficient of determination and the linear fit equation of the plotted data appear at the textbox and the legend respectively.

Table 3: WMO criteria compliance and correlation between CIMEL and PFRN27. $R^2$ is the coefficient of determination and 'Slope' corresponds to the linear fit of the CIMEL AOD monthly medians in relation to PFRN27 AOD monthly medians.

| CIMEL/PFRN27 comparison 2007-2019 | | | | |
|---|---|---|---|---|
| **Wavelength** | N | % within WMO limit | $R^2$ | Slope |
| 500/501 nm | 33197 | 95.56 | 0.987 | 1.003 |
| 870/862 nm | 33116 | 98.02 | 0.957 | 0.977 |

Most monthly AOD differences are within the monthly AOD median uncertainty (1σ) (the calculation procedure was described in Section 2.2.2) for all comparisons (Figure 4). The monthly AOD uncertainties vary for each month and dataset, with their

mean values being for the low frequency PFR dataset ($PFR_{syn}$) 0.0021/0.0017 in 501/862 nm and for the high frequency dataset ($PFR_{hf}$) 0.0008/0.0007 in 501/862 nm. In the CIMEL/PFR comparison the standard deviation of the monthly AOD differences is larger than the mean monthly AOD uncertainties, but at the same order of magnitude (0.0037/0.0028). The comparison between the 2 PFR datasets (low and high frequency $PFR_{syn}$/$PFR_{hf}$) shows that the measurement frequency differences can produce monthly AOD differences similar to those between CIMEL and PFR for synchronous datasets. The standard deviation

of the AOD differences (0.0034/0.0014 for 501/862) are larger than the $PFR_{syn}$ monthly AOD uncertainties at 500 nm (monthly uncertainty 0.0021) and lower at 862 nm (monthly uncertainty 0.0017) (Figure 4). In both wavelengths the standard deviation of the differences is larger for the CIMEL/PFR comparison where more monthly AOD differences exceed the monthly AOD 1σ uncertainty.




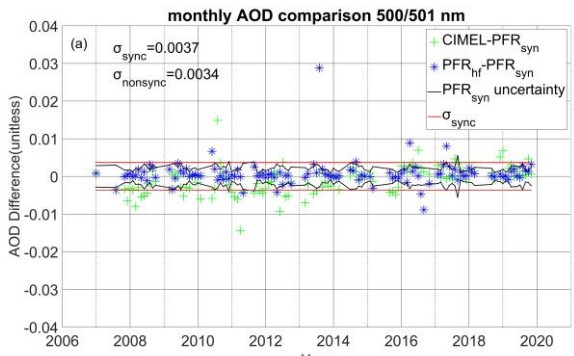

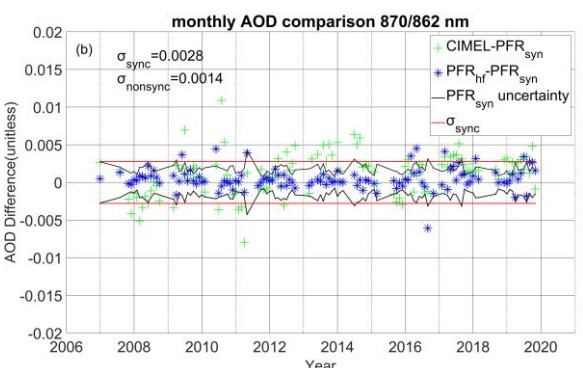

Figure 4: The monthly AOD differences for CIMEL/PFRN27 (green crosses) and the 2 PFRN27 datasets (blue stars) through the years for 500/501 nm (a) and 870/862 nm (b) with the uncertainties of the PFRN27 (synchronous with CIMEL) monthly AOD medians (black lines) and the standard deviation of the AOD differences between CIMEL and PFRN27 $\sigma_{sync}$ (red lines). The standard deviation of the $PFR_{hf}$-$PFR_{syn}$ differences appears in the upper left text as $\sigma_{nonsync}$.

## 3.2 Linear trends

### 3.2.1 Trend comparison on synchronous data

In this section we present the Davos AOD trends, assess the trend differences between the trends of monthly AOD calculated from the mean and the median of the measurements and between CIMEL and PFRN27 for the 2007-2019 period. For this period the AOD in Davos declined regardless the choice of instrument or averaging method. The CIMEL/PFRN27 trends derived from AOD monthly medians are -0.0129/-0.0178 per decade for 500/501 nm and -0.0048/-0.0074 per decade for 870/862 nm. The magnitude of the trends per decade correspond to 23.45%/31.79% of the mean AOD (0.055/0.056) for 500/501 nm and 19.20%/30.83% (mean AOD 0.024) for 870/862 nm. Figure 5 shows the datasets with the corresponding linear fitting and Table 5 includes the trends per decade and mean AOD values.

These AOD trends show a faster aerosol decline in comparison with previous studies about earlier periods. Specifically, in Ruckstuhl et al. (2008) there was a trend at 500 nm of -0.006 per decade for the period 1995-2005, but it was statistically insignificant. In Nyeki et al. (2012) the trend at 500 nm was positive (+0.002 per decade) with a mean AOD of 0.068 for the 1995-2010 period, but also not statistically significant.

The method of averaging systematically affects the trend per decade value and its statistical significance. At all datasets using the mean instead of the median results to a weaker trend, which is significant to a lower confidence level with the difference being up to approximately 10%. However, the effect is limited since all trend differences are smaller than the trend standard error (Table 4).

Table 4: Trends per decade calculated from monthly AOD means and medians for CIMEL and PFRN27 with their corresponding standard error and the p_values from the Mann-Kendall modified test (Section 2.2.2).





| | Median | | | Mean | | |
|---|---|---|---|---|---|---|
| | Trend per decade (X10⁻³) | Standard error (X10⁻³) | p-value | Trend per decade (X10⁻³) | Standard error (X10⁻³) | p-value |
| CIMEL 500 nm | -12.9 | 4.86 | 0.007 | -11.1 | 4.91 | 0.026 |
| PFRN27 501 nm | -17.8 | 4.92 | 0.000 | -15.2 | 4.87 | 0.002 |
| CIMEL 870 nm | -4.8 | 2.12 | 0.003 | -3.6 | 2.38 | 0.102 |
| PFRN27 862 nm | -7.4 | 2.19 | 0.000 | -6.8 | 2.34 | 0.106 |


Concerning the CIMEL/PFR trend comparison (calculated using the median hereafter), both instruments show a decline in AOD, which is statistically significant at higher than 97% confidence level. Trends and statistics for each dataset are presented in Table 5.

Despite the instruments' good agreement (Section 3.1) and the statistical significance of the individual trends, the linear trend
differences are not smaller than the trend standard error on all occasions. Specifically, at 865 nm the trend difference of $2.6X10^{-3}$ per decade is larger than the standard errors of the trends $2.12X10^{-3}/2.19X10^{-3}$. At 500 nm the trends differ by $4.9X10^{-3}$ per decade, while the standard error for the CIMEL trend is $4.86X10^{-3}$ and the PFRN27 trend $4.92X10^{-3}$ (Table 5). Also, the trend differences cannot be explained by the measurement uncertainty. The effect of the measurement uncertainty calculated from the Monte Carlo simulations at $1\sigma$ ($<6X10^{-4}$) is smaller than the trend differences ($>2.5X10^{-3}$) and the trend
standard error ($>2X10^{-3}$) for all time series.

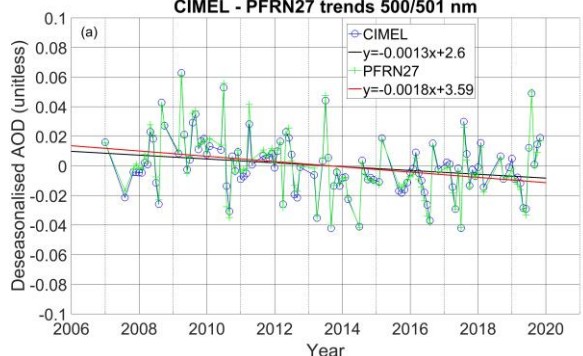

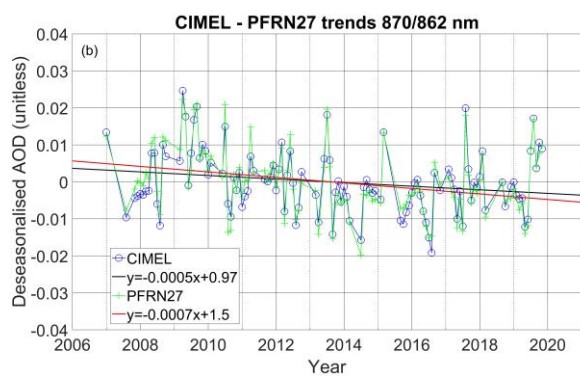

Figure 5: De-seasonalized AOD monthly medians (described in Section 2.2.2) and linear fits in 500/501 nm (a) and 870/862
nm (b). The blue circles correspond to the CIMEL monthly data and the green crosses to the synchronous PFN27. The black line is the linear fit result of CIMEL data and the red line the one for PFRN27. The linear fit equations appear in the legends.



Table 5: CIMEL/PFRN27 trends per decade comparison for synchronous datasets. The Monte Carlo trend standard deviation corresponds to the trend uncertainty attributed to the instrument measurement uncertainty.

| Time series | Trend per decade ($\times 10^{-3}$) | Standard error ($\times 10^{-3}$) | p-value observed | Monte Carlo trend st.d. ($\times 10^{-4}$) | Mean AOD |
|---|---|---|---|---|---|
| CIMEL 500 nm | -12.9 | 4.86 | 0.007 | 5.71 | 0.055 |
| PFRN27 501 nm | -17.8 | 4.92 | 0.000 | 5.69 | 0.056 |
| CIMEL 870 nm | -4.8 | 2.12 | 0.026 | 4.65 | 0.025 |
| PFRN27 862 nm | -7.4 | 2.19 | 0.000 | 4.49 | 0.024 |

### 3.2.2 Measurement frequency impact on AOD linear trends

In this section we compare the trends of the 2 PFR datasets (synchronous with CIMEL (syn) and high frequency(hf)). All trends are negative and statistically significant at higher than 99.99% confidence level (Table 6).

The trends in this case show better agreement than those of the previous section. The AOD trend differences due to different measurement frequencies are smaller than the trend differences between the CIMEL and PFR. Also, they are approximately 1 order of magnitude smaller than the trend standard error at both channels (Table 6).

The measurement uncertainty cannot explain these trend differences at 501 nm. At 862 nm it can be explained by the effect of the measurement uncertainty of the temporally low frequency dataset ($PFR_{syn}$). For this dataset the effect of the measurement uncertainty is larger due to the smaller number of measurements for each month.

Figure 6: Trend comparison between low (synchronous with CIMEL) and high frequency of measurements datasets.

| Time series | Trend per decade ($\times 10^{-3}$) | Standard error ($\times 10^{-3}$) | p-value observed | Monte Carlo trend st.d. ($\times 10^{-4}$) | Mean AOD |
|---|---|---|---|---|---|
| PFRN27 501nm (syn) | -17.8 | 4.92 | 0.000 | 5.69 | 0.056 |
| PFRN27 501nm (hf) | -17.2 | 5.00 | 0.000 | 1.98 | 0.057 |
| PFRN27 862nm (syn) | -7.4 | 2.19 | 0.000 | 4.49 | 0.024 |
| PFRN27 862nm (hf) | -7.2 | 2.21 | 0.000 | 1.78 | 0.024 |





### 3.3 Time-varying trends

#### 3.3.1 Synchronous time series

The DLM related analysis of trends for synchronous measurements appear in Figure 6. In contrast to the linear trends, for both instruments and wavelengths the trends are neither stable through the years nor monotonic. They are negative for the first 9

years approximately and positive for the rest of the period. The fact that the DLM trends are negative for most of the period is in line with the observed negative trends of the linear fitting method. The DLM trend uncertainties range extent to both negative and positive values for the whole period showing low significance in contradicting the very high significance of linear trends. The DLM trends show a better agreement between PFR and CIMEL compared to the linear trends, especially for the positive trend period. For both wavelengths the DLM trend differences between the instruments are clearly smaller than the trend

uncertainties ($1\sigma$). Also, the DLM trend comparison is consistent with the weaker linear trend of CIMEL compared to the PFR (Section 3.2.1). The DLM trends of CIMEL have lower absolute values for most of the years.

The linear trends are not fully consistent with the DLM trends. The DLM-linear trend differences for most years are larger than the linear trend standard error. On the contrary, the larger uncertainties calculated for the DLM trends through the MCMC method can explain all observed trend differences (Figure 6).


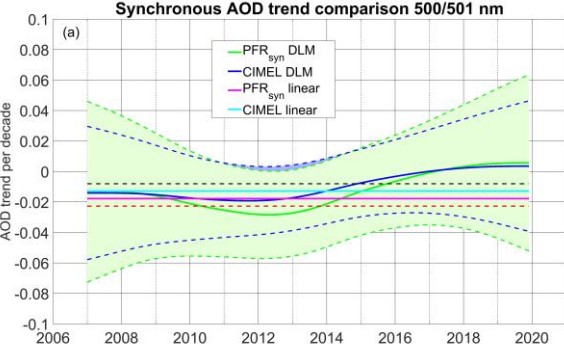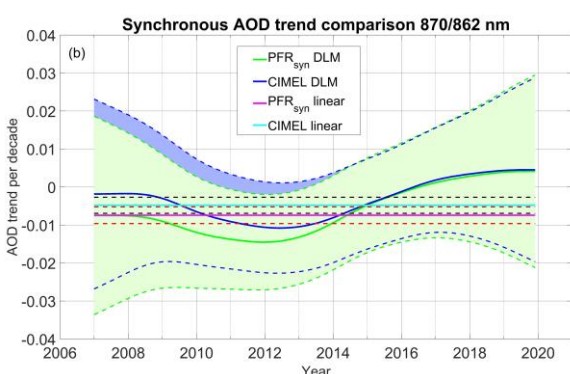

Figure 6: CIMEL/PFRN27 DLM and linear trends for 500/501 nm (a) and 870/862 nm (b). The green line is the PFRN27 DLM trend and the blue line the CIMEL DLM trend. The shaded areas show their uncertainty. The magenta line shows the linear trend for PFRN27 and the cyan for CIMEL, while the dashed red and black lines are the linear trend standard errors.


#### 3.3.2 Measurement frequency impact on AOD time-varying trends

The DLM trends for the 2 PFRN27 datasets (synchronous to CIMEL and high resolution) differ less than the CIMEL/PFRN27 trends, which is consistent with the linear trend comparison. Both trends are again well within the uncertainties and their differences are even smaller than the CIMEL-PFRN27 DLM differences in both wavelengths (Figure 7).





As was the case for the synchronous datasets, the linear trends differ with the DLM trends more than the linear trend standard
error for most years. The DLM trend uncertainties again are larger than all trend differences.

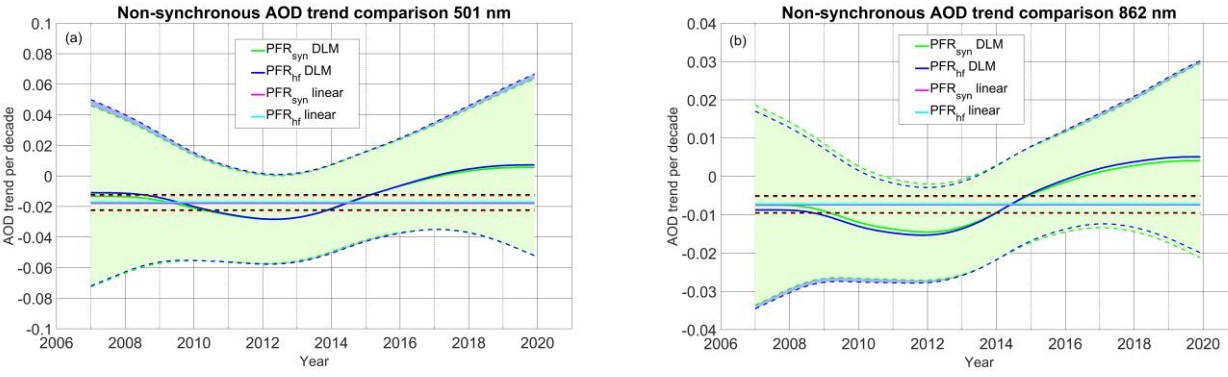

Figure 7: PFR$_{hf}$/PFR$_{syn}$ DLM and linear trends for 501 nm (a) and 862 nm (b). The green line is the PFR$_{syn}$ DLM trend and the
blue line the PFR$_{hf}$ DLM trend. The shaded areas show their uncertainty. The magenta line shows the linear trend for PFR$_{syn}$
and the cyan for PFR$_{hf}$, while the dashed red and black lines are the linear trend standard errors.

## 4 Summary and conclusions

In this study, we tried to take advantage of the 13-year period of AOD measurements from two different instruments belonging
to two different networks at Davos, Switzerland. We compared the time series between 2 different instruments measuring
AOD (CIMEL and PFRN27) for the period 2007-2019 regarding AOD measurement and trend differences in 2 channels
(500/501 nm and 870/862 nm). The instruments have different technical characteristics, cloud screening algorithms,
operational, calibration and processing protocols. The cloud screening algorithms agree for 93.8% of the coincident
measurements. The 2 instruments agree well according to the WMO criteria (>95% of synchronous AOD differences within
the WMO limits) for synchronous measurements and correlate well ($R^2$>0.95), while most of their AOD monthly median
values differ less than the monthly AOD uncertainty showing a very good consistency in calibration and post processing
methods. The selection of the averaging method (median instead of mean) affects the trends to an extent within the limits of
the standard error and increases their significance confidence level for up to approximately 10%. Because AOD does not
follow a normal distribution and the sensitivity of the median to outliers is lower, we consider the median a more representative
parameter for the monthly AODs.

The linear trends are higher compared to previous studies regardless the instrument selection, showing that there was an aerosol
load decline in Davos mainly after mid-2000s. The linear trends of both instruments under synchronous data are negative and
statistically significant at >97% confidence. However, their trend differences are large enough to equal or exceed the trend
standard error. The differences cannot be explained by the measurement uncertainty despite the small AOD in Davos (mean

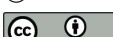



AOD 0.057 at 501 nm and 0.024 at 862 nm). Therefore, the measurement uncertainty of 0.01 used for calculating the monthly
AOD uncertainty does not induce a trend accuracy limitation large enough to enforce the observed trend differences. On the
other hand, assuming time-varying trends and using the Dynamic Linear Modeling, accounting for the uncertainties of the
monthly AODs, showed that the trend differences are smaller than the trend uncertainties.

The impact of measurement frequency on trends was explainable by the trend standard error and was found to be smaller than
the overall impact of the other instrument differences for the period 2007-2019. Our results suggest that the measurement
frequency differences between PFR and CIMEL do not affect the AOD trends significantly. However, different AOD absolute
values and variability compared to the ones in Davos, could enhance the impact of measurement frequency on AOD trends.
The effect can be also dependent on cloud screening algorithm differences. In our case the lower frequency dataset was filtered
with both GAW-PFR and AERONET cloud algorithms, while the high frequency dataset only with the GAW-PFR algorithm.
However, as mentioned earlier the 2 algorithms showed an agreement of 93.8% with this difference having very low effect on
AOD (less than 0.002/0.0005 for 501/862 nm). Also, the mean AOD for the high frequency PFR dataset is only 0.001 higher
than the mean AOD of the low frequency PFR dataset at 501 nm and 0.00 at 862 nm pointing to the short-term variability of
AOD as a main source of any monthly AOD differences caused by different measurement frequencies. Again, using the DLM
method the trend differences are well within the trend uncertainty.

Finally, the linear trends for the period 2007-2019 are not consistent with the DLM trends for the whole time period since the
latter are not monotonic. They are negative for the 2007-2016 period, followed by a short positive trend period. The linear
trend quantification cannot explain this inconsistency, while it is explained by the DLM uncertainties. Another inconsistency
is the high statistical significance of the linear trend that is not shown in the DLM trends.

The results of the paper cannot be used for any location and any instrument comparison but point out to the fact that when
calculating AOD trends a number of important factors including calibration coherency in time, post processing and cloud
elimination algorithm uncertainties, measurement frequency and even methods of AOD averaging or trend estimation should
be carefully considered.

*Code availability*. The Dynamic Linear Model code package is available from https://mjlaine.github.io/dlm/index.html

*Data availability*. The CIMEL AOD data are available from https://aeronet.gsfc.nasa.gov/

The PFR AOD data are available through communication with the authors.



*Author contribution*. AK analysed the data and wrote the paper with contributions from the co-authors. AK, SK conceptualized the study. NK, SK, JG contributed to the sun-photometer data provision. JG proposed and provided assistance with the DLM
trend analysis. LE provided assistance with the Monte Carlo simulations. All authors were involved in the interpretation of the results, reviewing and editing the paper.

*Competing interests*. The authors declare that they have no conflict of interest.

*Acknowledgments*. This work has been supported by the European Metrology Program for Innovation and Research (EMPIR)
within the joint research project EMPIR 19ENV04 MAPP "Metrology for aerosol optical properties". The EMPIR is jointly funded by the EMPIR participating countries within EURAMET and the European Union.






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
