# Peer review of "Sensitivity of aerosol optical depth trends using long term measurements of different sun-photometers"

_Atmospheric Measurement Techniques, 2022_

## Author Response (AR1)

**Author's response to the comment by Prof. Thomas Eck**

The authors would like to thank Prof. Thomas Eck for reading the manuscript and provide his helpful and constructive comment. You can find our response below the comment.

*Comment*

*I have a short comment related to the accuracy of the AOD analyzed in this paper. It is well known that sunphotometer measured AOD is proportional to the optical airmass (m) or pathlength through the atmosphere. The AOD error reduces by a factor of 1/m as m increases. This is reflected in your Figure 2 as the reduction in AOD differences between these two types of instruments as optical airmass increased. The most complete discussion of the accuracy of the AERONET measured AOD is given in Eck et al. (1999), where the uncertainty in measured AOD of field instruments is estimated to be 0.01 for airmass=1 (overhead sun) for visible and near-infrared wavelengths.*

*Therefore a potential additional analysis that could be added to this study to minimize the effects of calibration would be to utilize only data for m>3 for both instruments. Trends computed with this subset of data would therefore include only morning and afternoon data (excluding mid-day, although this would vary with season). In addition to a reduction of calibration biases between instruments by excluding mid-day data, there is also the added factor of excluding a significant portion of the mid-day data affected by fair weather cumulus clouds. All sunphotometer data sets are biased towards sampling low cloud fraction days with high atmospheric pressure. These days often show a diurnal cycle of cumulus cloud fraction related to the daily cycle of solar heating and associated convection and vertical mixing. Therefore an analysis of data with only m>3 or m>4 (in winter) would minimize the influence of a highly spatially and temporally variable cloud type on AOD (cloud edge contamination plus cloud influence of AOD itself; see Marshak et al., 2021), while also increasing AOD data accuracy. Of course the data sample size will decrease significantly for this large airmass subset of the data, but it should still provide for an additional informative aspect of this trend comparison for these two different instrument types which employ different measurement frequencies and cloud screening methodologies.*

Author's response

We added analysis for m>3 in sections 3.1 and 3.2.1 along with additions in the conclusions (section 4). To summarize, the AOD trend differences calculated for data with m>3 indeed were reduced compared to those calculated with all air masses and all differences are smaller than the trend standard error.

In more detail:

The data with optical air mass above 3 had some improvement in the AOD differences within the WMO limits (approximately 1% for 500/501 nm and 0.5% 870/862). In figure 1 below are the AOD differences and the WMO limits. There was also a small reduction to the standard deviation of the differences. The standard deviations are 0.0036 for 500/501 nm and 0.0026 for

870/862 nm. Using all air masses, the standard deviations are 0.0037 for 500/501 nm and 0.0028 for 870/862 nm.

[Figure]

Figure 1: CIMEL-PFRN27 differences (blue points and light blue to yellow bands) and WMO limits (red lines) of synchronous AOD measurements with respect to time in years for 500/501 nm (top) and 870/862 nm (bottom) for the PFR/CIMEL datasets with optical air mass above 3. The colourbar corresponds to the density of the AOD difference data points.

Regarding the monthly AOD medians, due to the data reduction the months we consider as valid reduced from 114 to 98. In order to compare the trends with and without air masses above 3 for the same months we re-calculated the trends from the full datasets (all air masses) using only these 98 months. On table 1 below we present the results of the trend analysis. Using only air masses above 3, we get smaller trend differences than the trend standard error and compared to the differences of the full datasets. The comparison without air mass restriction is consistent with the comparison present in the pre-print (114 months). Also, the trend difference due to the removal of 16 months is smaller than the trend difference between CIMEL and PFR. However,

the trend significance confidence level is reduced particularly for CIMEL if we use only 98 months.

Table 1: CIMEL/PFRN27 trends per decade comparison for synchronous datasets using only optical air masses above 3 and using all air masses for the common months.

| Optical air mass above 3 | | | | No optical air mass restriction | | | |
|---|---|---|---|---|---|---|---|
| Time series | Trend per decade($\times 10^{-3}$) | Standard error ($\times 10^{-3}$) | p-value observed | Mean AOD | Trend per decade($\times 10^{-3}$) | Standard error ($\times 10^{-3}$) | p-value observed | Mean AOD |
| CIMEL 500 nm | -15.8 | 6.41 | 0.008 | 0.058 | -11.7 | 5.11 | 0.032 | 0.057 |
| PFRN27 501 nm | -19.1 | 6.26 | 0.005 | 0.058 | -17.0 | 5.09 | 0.000 | 0.057 |
| CIMEL 870 nm | -4.9 | 2.67 | 0.119 | 0.025 | -4.1 | 2.24 | 0.134 | 0.025 |
| PFRN27 862 nm | -6.7 | 2.66 | 0.021 | 0.024 | -6.9 | 2.24 | 0.002 | 0.024 |

Author's changes in manuscript

In section 3.1 we added: '*The actual AOD uncertainty measured by CIMEL and PFR is a function of optical air mass (m) with the 0.01 value (section 2.2.2) corresponding to m=1 and reduces by a factor 1/m as m increases (Eck et. al., 1999; Kazadzis et al. 2020). This is evident in figure 2 where the AOD differences between CIMEl and PFR are reduced for higher air masses. In order to discuss on the effects of the calibration uncertainty to the calculated AODs we have used in a separate analysis only data for m>3 where the calibration effect on the AOD uncertainty is minimized. The number of measurements is 8304 for 500/501 and 8282 for 870/862 nm. The comparison of these data for coincident CIMEL and PFR showed 96.62% and 98.5% within the WMO limits and standard deviation of the differences 0.0036 and 0.0026 for 500/501 and 870/862 nm.*'

In section 3.2.1: '*Based on the high air mass analysis (m>3) described in section 3.1 we have calculated the trends for coincident PFR and CIMEL measurements. Because of the data reduction we removed some of the months used in the previous analysis creating additional data gaps. For m>3 the valid months are 98 instead of 114, which affects the trends. Therefore, we also re-calculated the trends shown in Table 5 (using all optical air masses) using only the 98 common months. The results of trend comparisons for all air masses and m>3 are in Table 6.*'

*Table 6: CIMEL/PFRN27 trends per decade comparison for synchronous datasets using only optical air masses above 3 and using all air masses for the common months.*' And the table present in this response.

In section 4 we added: '*In order to minimize the calibration uncertainty effects and reduce the AOD differences of the two different instruments we also compared their trends produced only from data with optical air mass above 3. The selection is based on the fact that the AOD uncertainty reduces for higher air masses. The trend agreement was improved as all trend differences are within the trend standard error.*'

**Author's response to referee #1**

The authors would like to thank the referee for providing helpful and constructive comments. You can find our response below each comment.

*Abstract: The three last sentences of the abstract seem confusing and don't help the reader to have an initial idea of the most relevant results of this work. One example can be found when the authors mention "all PFR data", or the final sentence about time-varying trends. I consider the information in the abstract should be self-explanatory for the reader to understand at a first glance the most important outcomes of the paper.*

Replaced the part: 'The trend differences are also larger than the trend uncertainty attributed to the instrument measurement uncertainty, with the exception of the comparison between the 2 PFR datasets (high and low frequency) at 862 nm. Finally, when calculating time-varying trends, they differ within their uncertainties.' with the following:

'Linear trend differences of the CIMEL and PFR time series presented here are not within the calculated trend uncertainties (based on measurement uncertainty) for 870/862 nm. On the contrary, PFR trends, when comparing high and low measurement frequency datasets are within such uncertainty estimation for both wavelengths. Finally, for time-varying trends all trend differences are well within the calculated trend uncertainties.'

*Page 2, line 46: Please include a more suitable reference for the Cimel sunphotometers, i.e., Holben et al. (1998) and/or Giles et al. (2019).*

Corrected as proposed.

The reference now is Holben et al. (1998).

*Page 2, line 48: Please include the word "SKYNET" in this sentence for clarification.*

Corrected as proposed.

The sentence now is: 'Most of the SKYNET sites are in East Asia and Europe.'

*Page 3, lines 85-87: It should be noted that this 13-year time series corresponds to Davos. Please also include a comma after "measurement frequency".*

Corrected as proposed.

*Page 4, line 123: Are the three Cimel sunphotometers used in this study calibrated in Mauna Loa?*

The calibration of CIMEL-sunphotometers at AERONET-Europe started in 2006 (since 2011 within the ACTRIS framework), therefore all presented data have this type of calibration.

*Page 4, line 24: There is a typo in Giles et al. (2019).*

Corrected.

Now it is Giles et al. (2019) instead of Gilles et al. (2019).

*Page 5, lines 140-142: The authors enumerated in lines 70-73 different references in the literature aimed at inducing a threshold in the minimum amount of daily/monthly information to reliably perform statistical analysis. However, the authors finally set the threshold at 5 (3) to have a valid month (day) median. Is this threshold an empirical output of this specific study? Is this a recommendation for future studies?*

The definition of such thresholds is empirical and is always a compromise between data availability and representativeness of the final statistics. For the case of low measurement frequency instruments (e.g. CIMEL) this compromise becomes more difficult, especially at locations with frequent cloudiness. For the case of the PFR, the number of measurements are 10 to 15 times higher, so also higher thresholds (higher number of minutes per day and days per month) could be used in order to improve the quality of the statistics on calculated medians and trends. So, the recommendation is for CIMEL or PFR users to adjust such thresholds empirically based on tests considering their data availability and the cloud conditions for each location under study.

*Page 5, lines 151-153: The authors found 93.8% of Cimel-synchronous data to be cloud-free according to the PFR cloud screening algorithm. However, I don't understand the next step. Do the authors calculate the AOD average (not the median value) of the complete data series and compare this value with the average of 93.8% of data? Please clarify.*

Yes, we compare the 2 averages (mean value) of all the measurements of each dataset (100% of the data and 93.8% of the data). Replaced the word 'average' with 'mean' in the text and added a clarification sentence.

The full sentence now with the added parts in italics is: 'Keeping only this 93.8% of CIMEL and PFR synchronous data reduced the average *mean* AOD of both instruments by less than 0.002 at 500/501 nm and less than 0.0005 at 870/862 nm *compared to the mean AOD of 100% of the data*, pointing towards the conclusion that the cloud contamination effects on AOD are minimal.'

*Page 5, last sentence: The location of this sentence at the end of the section seems confusing. Do the authors have any reason to have placed it at the end of the section? Is it possible to include this information in Table 1?*

Moved the paragraph earlier in the same section. Now it starts at line 144 of the preprint.

Added the number of months to table 1.

*Page 6, line 163: Can the authors explain briefly the way to de-seasonalize monthly medians?*

We calculated the mean value of all medians for each month (intra-annual cycle) and subtracted it from each of the monthly medians. Added clarification in the same sentence.

The full sentence now with the added parts in italics is: 'To *calculate* the *de-seasonalized monthly medians, the* intra-annual cycle was calculated separately for each dataset from all medians for each month *and subtracted from each monthly median*.'

*Page 7, line 196: Do the authors have any explanation for the different median/mean AODs in June at your site? Maybe is it related to the arrival of different air masses and therefore some outliers are likely to occur at this time of year?*

On general the warm months, which are the months with higher AOD, show larger mean/median difference compared to cold ones. The difference between June and the other warm months like July is the very large fraction of June measurements below AOD 0.06 compared to the nearby higher values (0.06-0.1) and a larger fraction of exceptionally high values e.g. AOD above 0.3), which lead to a higher deviation of AOD values from a normal distribution during June. A possible explanation for increased high AOD events could be the presence of more Sahara dust events during these 2 months is not consistent with the 2001-2011 mean and 2012 for the Swiss Alps according to meteoswiss: https://www.meteoswiss.admin.ch/home/climate/the-climate-of-switzerland/specialties-of-the-swiss-climate/saharan-dust-events.html
It is likely that during June the intrusions of unusually large aerosol quantities from the surrounding areas (e.g. combustion products) are more frequent. This could be linked to the high heating rate of the ground during early summer that causes high atmospheric vertical instability hence more effective aerosol transport and to the wind patterns.

*Page 8, line 207: Similarly to the last question, a clear departure of AOD differences from the WMO criterion of traceability is observed in 2019. Do you have any clue to explain this unexpected behaviour?*

This is a case where it looks visually like there is a particular departure of AOD differences compared to the other years, but this is not true. 2019 is the year with the most measurements. The larger amount of data outside WMO limits are not a larger fraction of the data compared to all other years or a big departure from the traceability criterion (94.68% of the year's data are within the limits at 500 nm). In terms of data percentage within the limits, standard deviation of differences and 95th percentile-5th percentile difference, it is similar with other years.

*Page 9, Figure 3 caption and Table 3 caption: coefficient of determination is sometimes called as an acronym and sometimes not. Please homogenize.*

Corrected.

The figure 3 caption now with the added part in italics is: The coefficient of determination *($R^2$)* and the linear fit equation of the plotted data appear at the textbox and the legend respectively.

*Page 11, Figure 5: Y-axis should be "de-seasonalized", according to the text and figure caption.*

Corrected.

*Page 13, line 307: Is there an objective threshold for statistical significance in the DLM analysis? What does "low significance" mean?*

The trend is considered statistically insignificant when the trend uncertainty includes the null hypothesis (zero trend).

Replaced 'low significance' with 'lack of significance for most months'.

*Page 13, section 3.1.1 and conclusions: The fact that LSLR trends are not consistent (in terms of tendency and significance) with DLM trends makes the reader ask him/herself about the suitability of including this analysis. The different tendencies can be certainly explained because of the fact that trends are not monotonic, as it is stated in the paper, and this type of analysis (DLM) seems therefore more adequate to study the long-term fluctuation of a variable like AOD. However, DLM results lack of statistical significance. Have the authors checked the existence of some break-points in the de-seasonalized monthly AOD data series in Fig. 5? Looking at these pictures, a real change in the trend during the last part of the period could be discerned, which could add weight to the positive trend found in the DLM analysis.*

Indeed there is a visible increase at the end of the period. However, choosing visually break-points can be challenging and subjective. The DLM analysis provides a more objective criterion to spot break-points. According to this analysis we could separate the LSLR trends in one negative trend period with length of 8.5 years and a positive trend period of 4.5 years. If we do this, we get as expected stronger negative trends the first 8.5 years compared to the 13-year trends (with a larger standard error) and positive for the last 4.5 years (with even larger standard error). The confidence level of significance of the negative trends vary between 96.5% and 99.77% and for the positive trends between 87.62% and 99.91% depending on the timeseries and the wavelength. However, calculating the LSLR in these 2 periods does not overturn any of the present conclusions. For example, during the negative trend period the trend difference between CIMEL and PFR at 870/862 nm is still larger than the trend standard error and the effect of the measurement frequency smaller. Therefore, we had decided that it was not necessary to include this analysis on the final paper.

Our goal was to assess the trend differences between different instruments, measurement frequencies and methods. The inclusion of both methods show not only the different outcome from each approach, but also how datasets from different instruments or with different measurement frequencies compare separately for each method.

*Pages 14-15, conclusions: In general, this section seems confusing for the reader, with some redundant information that might be removed. Some examples at page 15, line 359 ("as mentioned earlier") or page 15, line 363 (redundant sentence about the trend uncertainty). Furthermore, the time sequence of the writing does not correspond to the timeline of this paper.*

Removed the 2 sentences and re-wrote parts of the section to be easier for the reader and follow the timeline of the paper. Replaced 'linear' with 'LSLR'

The changed parts in the conclusion section with the added words in italics are:

Starting at l. 338 of the pre-print: 'The 2 instruments agree well *on AOD measurements* according to the WMO criteria (>95% of synchronous AOD differences within the WMO limits) for synchronous measurements.'

Starting at l. 340 of the pre-print: '*Because AOD does not follow a normal distribution we compared the intra-annual cycles calculated by either mean monthly and median monthly values. We decided to use the medians as monthly AOD, because the sensitivity of the median to outliers is lower and we consider it a more representative parameter for our data.*

*The monthly median AOD values of the 2 instruments* correlate well ($R^2$>0.95) and most of their AOD monthly median values differ less than the monthly AOD uncertainty showing a very good consistency in calibration and post processing methods.

*We performed a set of different trend analyses corresponding to the study's goals. Firstly, we compared least squares linear regression (LSLR) trends using de-seasonalised monthly means and de-seasonalised monthly medians to investigate the sensitivity of trends on the method of averaging.* The selection of the averaging method *affected* the trends to an extent within the limits of the standard error. *The selection of medians instead of means increases increased the trend* significance confidence level for up to approximately 10%*. Only the monthly medians were used for the rest of the trend comparisons.*

The *LSLR* trends *in this study* are higher compared to previous studies regardless the instrument selection, showing that there was an aerosol load decline in Davos mainly after mid-2000s. The *LSLR* trends of *CIMEL and PFR* instruments under synchronous data are negative and statistically significant at >97% confidence. However, their trend differences are large enough to equal or exceed the trend standard error. *Another source of trend uncertainty is the measurement uncertainty. Using the Monte Carlo method to quantify the trend uncertainties due to a measurement uncertainty of 0.01, it is evident that the* differences cannot be explained by *this* uncertainty despite the small AOD in Davos (mean AOD 0.057 at 501 nm and 0.024 at 862 nm).

*In order to minimize the calibration uncertainty effects and reduce the AOD differences of the two different instruments we also compared their trends produced only from data with optical air mass above 3. The selection is based on the fact that the AOD uncertainty reduces for higher air masses. The trend agreement was improved as all trend differences are within the trend standard error.'*

Starting at l. 364 of the pre-print: 'Finally, *we used Dynamic Linear Modeling (DLM) to estimate time-varying trends. In this case, the trend comparison between CIMEL and PFR is improved as all trend differences are smaller than the trend uncertainties. On the other hand, the comparison between linear and DLM trends shows some important differences.'*

*Page 14, line 346: What do the authors mean by a decline after mid-2000? Linear trends are observed to be negative in the whole 2007-2019 period.*

This is a reference to the combination of the previous findings and this study (section 3.2.1). Ruckstuhl et al. (2008) and Nyeki et al. (2012) found statistically not significant negative and positive trends for the periods 1995-2005 and 1995-2010 respectively, while we found statistically significant negative trend starting 2 years after 2005.

*Page 15, line 351: Dynamic Linear Modeling does not appear with the acronym, while this section includes many of them (including also DLM in lines 362, 364, 366 and 367).*

Added the acronym after the phrase.

*Page 15, line 361: Please correct the typo "0.00 at 862 nm"*

The difference is 0 at 3-digit accuracy, which we used. Added another 0.

**Author's response to referee #2**

The authors would like to thank the referee for providing helpful and constructive comments. You can find our response below each comment.

*Explain in the text, for who is not of this field, the level of AOD aeronet downloaded data*

Added the part below as explanation at the end of section 2.1.2.

'The CIMEL AOD data are publicly available at 3 levels. Level 1.0 are near real-time data without cloud screening, the final calibration and quality assurance. The cloud screening produces the level 1.5 data also near real-time. After the application of the final calibration and quality assurance the level 2.0 data are produced, which we use in this study.'

*Line 145, in this part of the text it is not clear what the PFRhf dataset is used for, even if explained clearly later.*

Added a clarification in the sentence.

The sentence now is the following with the added part in italics: 'The second one, $PFR_{hf}$, is a much larger dataset that represents the PFR measurement frequency (1 min) *and its comparison with $PFR_{syn}$ can show the effect of the measurement frequency on AOD differences and trends*.'

*Line 207: any reason for the larger deviation in 2019?*

This is a case where it looks visually like there is a larger deviation of AOD differences compared to the other years, but this is not true. 2019 is the year with the most measurements. The larger amount of data outside WMO limits are not a larger fraction of the data compared to all other years or a big departure from the traceability criterion (94.68% of the year's data are within the limits at 500 nm). In terms of data percentage within the limits, standard deviation of differences and 95th percentile-5th percentile difference, it is similar with other years.

**Other changes**

We added few additional changes. Some as a result to the received comments, others due to mistakes spotted and improvements decided after the pre-print publication.

1. Homogenized the wavelengths in the text by replacing any 500 nm with 500/501 nm and 865 nm with 870/862 nm.

2. l. 89 of the revised version with tracked changes added a clarification(italics): 'For this study, we use 13 years of parallel PFR and CIMEL timeseries *at Davos, Switzerland*'

3. Added more information in the caption of Table 1 (italics): The number of measurements *of* the datasets *(before removing 'invalid' months)*, the time period used *and the number of months considered as valid. Keeping only the valid months reduced the overall number of measurements by 3.5%.*

4. Former table 6 is now table 7.

5. Added in the references: 'Eck, T. F., Holben, B. N., Reid, J. S., Dubovik, O., Smirnov, A., O'Neill, N. T., Slutsker, I. and Kinne, S.: Wavelength dependence 460 of the optical depth of biomass burning, urban, and desert dust aerosols, J GEOPHYS RES-ATMOS, 104(D24), 31333-31349, https://doi.org/10.1029/1999JD900923, 1999.'